# The National Pharmaceutical Council: Endorsing the Construction of Imaginary Worlds in Health Technology Assessment

**DOI:** 10.3390/pharmacy8030119

**Published:** 2020-07-13

**Authors:** Paul C Langley

**Affiliations:** College of Pharmacy, University of Minnesota, 308 SE Harvard St, Minneapolis, MN 55455, USA; langley@maimonresearch.com

**Keywords:** National Pharmaceutical Council, imaginary value assessments, mathematically impossible QALYs, Rasch measurement

## Abstract

All too often, organizations embrace standards for health technology assessment that fail to meet those of normal science. A value assessment framework has been endorsed that is patently in the realm of pseudoscience. If a value assessment framework is to be accepted, then claims for the value of competing products must be credible, evaluable and replicable. If not, for example, when the assessment relies on the construction of an imaginary lifetime incremental cost-per-quality-adjusted-life-year (QALY) world, then that assessment should be rejected. Such an assessment would fail one of the central roles of normal science: the discovery of new facts through an ongoing process of conjecture and refutation where provisional claims can be continually challenged. It is no good defending an endorsement of a value framework that fails expected standards on the grounds that it has been endorsed by professional groups and reflects decades of development. This is intellectually lazy. If this is the case, then the scientific revolution of the 17th century need not have happened. The purpose of this commentary is to consider the recommended standards for health technology assessment of the National Pharmaceutical Council (NPC), with particular reference to proposed methodological standards in value assessment and the commitment to mathematically impossible QALYs.

## 1. Introduction

One feature that sets health technology assessment apart from the other social sciences, including mainstream economic analysis, is the commitment to the construction of incremental cost-per-quality-adjusted life year (QALY) imaginary worlds to support competing value claims for products and devices. This is an absurd position, but one that is rigorously supported by the leaders in the field of cost-effectiveness analysis [1]. For those who have been trained in the standards of positive economics, the role that is assigned to the discovery of new facts, theory construction and hypothesis testing, this focus on imaginary lifetime incremental cost-per-quality adjusted life year (QALY) worlds, and their wholehearted embrace by organizations such as the International Society for Pharmacoeconomics and Outcomes Research (ISPOR) and groups such as the Institute for Clinical and Economic Review (ICER) in the US, is absurd. The active pursuit of approximate information and the rejection of hypothesis testing to support formulary decision assessment is an analytical dead end, a feature that was recognized 30 years ago [2]. 

The purpose of this commentary is to review the recommended health technology assessment methodology and standards of the US National Pharmaceutical Council (NPC) when set against the standards of normal science [3] all claims made for competing pharmaceutical products and devices must be credible, evaluable and replicable. These are unexceptionable standards. They point to, at least in the physical sciences, the importance of instrument development and measurement, with a commitment to empirical testing. This commitment to empirical assessment, the discovery of provisional new facts, sets normal science in opposition to pseudoscience. In health technology assessment, the construction of imaginary worlds puts these activities squarely in the realm of pseudoscience, sharing the Dover courtroom with intelligent design rather than the proponents of natural selection [4].

It is also important to reject, as a key element in this neglect of the standards of normal science, the acceptance of lifetime models, in particular the QALY model recommended by ISPOR as the basis for providing approximate information to formulary decision makers. The problem here, other than the absurdity of lifetime models, is that the QALY is a mathematically impossible construct. It relies on a failure to understand, or more possibly ignore, the axioms of fundamental measurement. As detailed below, utilities are ordinal scales; they cannot be applied to create QALYs.

## 2. The National Pharmaceutical Council’s Position

The NPC recommendations appear to conflate real world evidence with the creation of imaginary world evidence, with NPC’s commitment to value assessment clearly in the latter, lifetime imaginary worlds, camp. While the NPC asks us to focus broadly on all aspects of the health system not just on medications in value assessment, which presumably would include real-world evidence as well as modeled imaginary claims for value in target populations, there is little guidance that would assist manufacturers in making claims. The weakness in the NPC position is the failure to reject imaginary constructs, to provide a coherent counter narrative. The weak NPC position in imaginary value assessment can be addressed from a number of perspectives:Neglect of the standards of normal science, a failure to distinguish normal science from pseudoscience and acceptance of the view that evidence is created not discovered with a commitment to the ISPOR position that consensus is truth; the creation of approximate imaginary information;A commitment to the role of lifetime frameworks in the creation of imaginary information, which fails standards for credibility, evaluation and replication;A belief that it is possible to identify realistic assumptions that will hold over the time horizon of the imaginary world where “realistic assumptions” create “realistic claims”, although these claims can never be evaluated empirically;A belief in the ability to create claims for the future from observations about the past; a commitment to inductivist philosophy (logical positivism), which was overturned some 80 years ago;To support, following NPC’s commitment to following accepted methodological standards, lifetime incremental cost-per-QALY models where the QALY is a mathematically impossible construct, which leads to a manifest failure to appreciate the importance of instruments for claims evaluation that are driven by the axioms of fundamental measurement.

## 3. Standards of Normal Science

It is important for organizations such as the NPC to understand the basis on which new evidence is provisionally discovered (not proved). The paradigm that supports discovery in the development of pharmaceutical products through the phases of drug development should apply equally to claims for the impact of products in treating populations. We do not ask manufacturers to create evidence from assumptions; the evidence will emerge from a process of conjecture and refutation or hypothesis testing. If the evidence to support claims is not available at product launch, then instead of creating imaginary cost-utility constructs to generate ersatz evidence claims, the focus should be on evidence platforms to support models with credible and evaluable claims.

The requirement for testable hypotheses in the evaluation and provisional acceptance of claims made for pharmaceutical products and devices is unexceptional. Since the 17th century, it has been accepted that if a research agenda is to advance, if there is to be an accretion of knowledge, there has to be a process of discovering new facts. NPC appears to be opposed to this. By the 1660s, the scientific method, following the seminal contributions of Bacon, Galileo, Huygens and Boyle, had been clearly articulated by associations such as the Academia del Cimento in Florence (1657) and the Royal Society in England (founded 1660; Royal Charter 1662) with their respective mottos *Provando e Riprovando* (prove and again prove) and *nullius in verba* (take no man’s word for it) [5]. 

By the early 20th century, standards for empirical assessment were put on a sound methodological basis by Popper (Sir Karl Popper 1902–1994) in his advocacy of a process of “conjecture and refutation” [6,7]. Hypotheses or claims must be capable of falsification; indeed, they should be framed in such a way that makes falsification likely. Although Popper’s view on what demarcates science (e.g., natural selection) from pseudoscience (e.g., intelligent design) is now seen as an oversimplification involving more than just the criteria of falsification, the demarcation problem remains [8] Certainly, there are different ways of doing science, but what all scientific inquiry has in common is the “construction of empirically verifiable theories and hypotheses”. Empirical testability is the “one major characteristic distinguishing science from pseudoscience”; theories must be tested against data. Hence, pivotal clinical trials, not simulated imaginary worlds with selected data inputs from pivotal trial data to recycle old (and imagined) facts. We can only justify our preference for a theory by continued evaluation and replication of claims. Constructing imaginary worlds, even if the justification is that they are “for information”, is, to use Bentham’s (Jeremy Bentham 1748–1832) memorable phrase, “nonsense on stilts”. If there is a belief, as subscribed to by the NPC, in the sure and certain hope of constructing imaginary worlds, to drive formulary and pricing decisions, then it needs to be made clear that it is a belief that lacks scientific merit. It fails the demarcation test; it is pseudoscience (i.e., pure bunk). It is difficult to judge whether or not the NPC is aware of the standards of normal science—activities that distinguish natural selection from intelligent design. It is important, given NPC’s position as a national opinion center, with its contribution to policy setting, for this to be made clear.

## 4. Approximate Information (or Disinformation)

It is worth emphasizing that ISPOR, as noted above, ICER’s methodological mentor, explicitly disavows hypothesis testing as a core activity in health technology assessment. NPC presumably concurs. The primary role of health technology assessment for ISPOR (and ICER) is to create “approximate information”. It is not clear what this means (presumably it can be distinguished from “approximate disinformation”) as there is not, in the imaginary world of ISPOR and ICER modeling, any known reference point for “true information” to judge approximation. How close are we? It is difficult to be approximate to the “truth” when the context is imaginary and the “truth” will only be revealed 10, 20 or 30 years or more ahead if all the assumptions in the model are realized. The Oxford English Dictionary definition may relate approximately to a “known” truth, but in the construction of imaginary worlds there can be no such reference point.

It is difficult to judge how formulary committees would react to the NPC saying it supported the construction of approximate and unevaluable “approximate information” in decisions. Does imaginary evidence (or claims) constructed from lifetime models necessarily supplant evidence-based medicine where claims are provisional and where a research program could be proposed to capture evidence that was unavailable at product launch? Constructing imaginary evidence is, of course, a lot easier and more cost-effective; with the opportunity, as noted by ICER, to revisit imaginary claims if data become available to modify assumptions, producing a new round of imaginary and nonevaluable claims. Proposing a research program does not mean waiting for decades. When manufacturers make a claim, they should be asked to present it as a protocol that can propose feedback in a meaningful time frame to formulary committees.

## 5. Choice of Assumptions

One of the more intriguing elements in the NPC recommendations is the insistence on “realistic assumptions”. But what does this mean? Is there an accepted distinction, a criterion for categorizing assumptions as “realistic”? Is it possible to be unequivocal as to the realism of assumption that might hold over the lifetime of modelled target patient populations? Presumably NPC joins with ISPOR and ICER in the fabrication of imaginary worlds that set the stage for value impact over 10, 20 or 30 years. Typically, the number of assumptions that have been assembled to support the various simulations and their scenario progeny across therapies can be truly awesome; some come from the literature, others are pure guesswork. This does not mean there is only one possible model; there is presumably scope for a multiverse of models, each with their own family of scenarios, each producing claims that can never be evaluated. Indeed, they were never meant to be capable of evaluation. That is the great advantage of building assumption-driven imaginary worlds; only the assumptions can be challenged (which seems a fruitless endeavor). Is there a proposed NPC reference case model?

Unfortunately, even if an assumption driving the imaginary value assessment framework is defended by appealing to the literature (including pivotal clinical trials) the effort is wasted. The point, and this goes back to Hume’s (David Hume 1711–1776) induction problem, is that we cannot ask clients in health care to believe in models constructed on the belief that prior assumptions will hold into the future. It is logically indefensible: it cannot be “established by logical argument, since from the fact that all past futures have resembled past pasts, it does not follow that all future futures will resemble future pasts” [9]. 

## 6. Utilities and QALYs

QALYS are the Achilles heel of the ISPOR and ICER model universe. Exeunt QALYs and the fantasy edifice collapses. Apart from their use in the ISPOR and ICER contribution to the science fiction literature, QALYs can only survive if the measure is credible, evaluable and replicable. The concept of a QALY is not new; it goes back some 40-plus years with the notion of combining time spent in a disease state with some multiplicative “score” on a required ratio scale of 0 to 1 (death to perfect health). Combining the two, multiplying time by utility is assumed to produce a QALY. However, before considering the utility that is central to the imaginary ICER simulation, a brief digression on measurement theory and its application to instrument development in the social sciences is in order. There are four main types of measurement scale, putting to one side conjoint simultaneous measurement, which underpins Rasch Measurement Theory (RMT) [10]. These are nominal, ordinal, interval and ratio. Each satisfies one or more of the properties of (i) identity, where each value has a unique meaning; (ii) magnitude, where each value has an ordered relationship to other values; (iii) interval, where scale units are equal to one another; and (iv) ratio, where there is a “true zero” below which no value exists. Nominal scales are purely descriptive and have no inherent value in terms of magnitude. Ordinal scales have both identity and magnitude in an ordered relation, but the unknown distances between the ranks means the scale is capable only of generating medians and modes and the application of nonparametric statistics. The interval scale has identity, magnitude and equal intervals. It supports the mathematical operations of addition and subtraction. A ratio scale satisfies all properties, supporting the additional mathematical operations of multiplication and division. Recognition and adherence to these fundamental axioms of measurement theory is critical if a measure is to have any credibility. In the physical sciences, this has been long recognized; accurate measurement is the key to hypothesis testing and the discovery of new facts. The same arguments apply to the social sciences. Unfortunately, they appear all too often to be absent in health technology assessment.

The case presented here is that the EQ-5D-3L and other multiattribute instruments generate ordinal scores [11]. It does not have interval properties (i.e., invariance of comparisons); neither does it have ratio properties, as the EQ-5D-3L “score” lacks a true zero (i.e., distance from zero). Of course, if the EQ-5D-3L fails to demonstrate interval properties, then it is a waste of time to consider whether it has ratio properties. The actual range for the EQ-5D-3L is not from 0 = death to 1 = perfect health, but from −0.59 to 1.0; the algorithm to compute utilities allows negative values (and always will). The fact that the EQ-5D-3L has ordinal properties is easily demonstrated: the symptom elements that comprise the EQ-5D-3L attributes are on ordinal scales. Simply applying community preference weights and adding these up still results in an ordinal scale.

There is the further question of unidimensionality. Measurement scales should have the property of unidimensionality. The focus should be on one attribute at a time. We must avoid confusing a number of attributes into a single score. Mutiattribute scales such as the EQ-5D-3L reduce confidence in predictions, and the score is a less useful ordinal summary. In Rasch modeling, estimates of item difficulty and person ability are meaningful if every question contributes to the measurement of a single underlying attribute. Our analytical procedures, if we are to meet the property of unidimensionality, must incorporate indicators of the extent to which the persons and items fit our concept of an ideal unidimensional line. Items should contribute in a meaningful way to the construct/concept being investigated.

In the case of the EQ-5D-3L and other multiattribute scales, the notion of unidimensionality is absent. While it is claimed to capture health-related quality of life (HRQoL), there is no single attribute or latent construct. It comprises five symptoms (mobility, self-care, usual activity, pain/discomfort, and anxiety/depression) with three ordinal response levels (no problem, some problems and major problems), creating a multiattribute scale with ordinal properties. Each of the symptoms is an attribute that could be the foundation for its own unidimensional scale. While ISPOR and ICER apparently believe the EQ-5D-3L has ratio properties, this, as noted, is demonstrably false given negative utilities. However, perhaps this is not as egregious as the “false assumption” position taken by authors, where it is acknowledged that the EQ-5D-3L lacks a true zero but that, in order to maintain the QALY illusion, we assume it has ratio properties [12]. 

The situation does not change when we move from the EQ-5D-3L to the EQ-5D-5L (introduced in 2009), where there are five response levels. Increasing the allowed ordinal responses to five reduces the number of respondents with extreme problems. The result is a range, still including negative utilities, from −0.29 to 1.0. Even so, it is still an ordinal score.

Even if ISPOR and ICER were willing to recognize the absence of fundamental measurement properties in the EQ-5D-3L (and other generic utility instruments), this does not mean that this would give succor to the belief in fabricated imaginary evidence. The ICER value assessment framework would still fail the demarcation test as pseudoscience. It is also difficult to see how ICER might underwrite a “utility” instrument that met the standards required (a true zero yet capped at unity). After all, instruments developed by the application of RMT focus, as noted below, on the response to interventions on a constructed interval scale from ordinal responses rather than attempting to go the further step of creating instruments that have ratio properties [13,14,15]. 

## 7. A Way Out: Rasch Measurement Theory

The ISPOR and ICER value assessment framework, as noted above, is an analytical dead end [16] This has been known for some time, yet organizations continue their financial support of ICER and its construction of imaginary value assessments. While it is not the purpose here to question these decisions, there is a way out. This involves a commitment to the standards of normal science and the primacy of real-world evidence. If this commitment is made, then the imaginary value assessment, creating approximate “pseudo realistic” information, can be abandoned along with the belief in the existence of a true zero for generic multiattribute utility scales. 

The more substantive step is to buy into the axioms of fundamental measurement and, following the physical sciences, commit to constructing instruments to evaluate quality of life that meets Rasch measurement standards. This provides a focus for constructing interval scale response measures that are disease-specific and consider quality of life from a patient needs fulfillment perspective. Rasch measurement theory (RMT) is not compatible with either classical test theory (CTT) or item response theory (IRT) They are, as Bond and Cox point out, competing paradigms^10^ RMT takes the perspective that if the instrument is to meet fundamental measurement standards, then we should adopt the Rasch data-to-model paradigm. If we are not concerned with, or are happy to ignore, questions of fundamental measurement, then we can follow the CTT or IRT model-to-data paradigm. The key distinction is that *−RMT* uses the measurement procedures of the physical sciences as the reference point^108^. We can aim for the standards in the physical sciences by, as Stevens pointed out in the 1940s, allocating numbers to events according to certain rules [17] It is these rules that comprise RMT. To reiterate: RMT is designed to construct fundamental measures. CTT and IRT focus on the observed data; these data have primacy and the results describe those data. As Bond and Cox emphasize: in general, CTT and IRT are exploratory and descriptive models; the Rasch model is confirmatory and predictive^10^. If RMT is ignored then, by default, instruments utilizing Likert scales or similar frameworks will fail to meet the required axioms of fundamental measurement and remain ordinal scales.

A commitment to disease-specific, patient-centric interval response instruments provides a firm foundation for evidence-based medicine. A number of these have been developed following RMT standards; one example is the Crohn’s Life Impact Questionnaire (CLIQ) [18,19]. We can abandon imaginary lifetime value assessments and focus instead on claims for quality of life that are credible, evaluable and replicable. We can focus on discovering new facts rather than recycling assumptions. It is unlikely, however, that a positive outcome as outlined above will have any chance of mainstream success. Health technology assessment has far too much to lose. Leaders in organizations such as ISPOR have invested 30 years of academic and pseudo-academic endeavor into constructing imaginary worlds and proposing the dominant role of approximate information or disinformation in decision making. Indeed, as the QALY is an impossible construct then we might more appropriately discard the notion of approximate information and use instead the term impossible information. Can the NPC make a difference?

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
