# Peer review of "The National Pharmaceutical Council: Endorsing the Construction of Imaginary Worlds in Health Technology Assessment"

_pharmacy, 2020, doi:10.3390/pharmacy8030119_

Round 1
Reviewer 1 Report
This was a provocative commentary paper espousing that HTA is pseudoscience and the “QALY is a mathematically impossible construct”. There are several fonts and sizes used in the article.
Line 34 The author should define what he means by HTA and ‘value assessment’ plus the role of the NPC (there is potentially an international audience). The author used the term HTA but I wonder if he really means economic evaluations and, more specifically, cost utility analyses. HTA encompasses aspects well beyond a cost-utility analysis. The WHO also supports HTA (not just ISPOR and ICER): https://www.who.int/medical_devices/assessment/en/
In recent months, many governments have relied on modelling to make decisions about how to manage the COVID-19 pandemic. Would the author argue that such ‘imaginary world’ modelling is also ‘pseudoscience’?
Line 42 I am not sure of their reference to the ‘Dover courtroom’. Needs a citation.
Line 90. Presumably ICER here means the organisation and not the metric.
Line 124 define OED
Lie 27. The author proposes a ‘way out’ but the actual implementation of the proposal is not clear. How would the RMT be manifested in practice?
Author Response
This was a provocative commentary paper espousing that HTA is pseudoscience and the “QALY is a mathematically impossible construct”. There are several fonts and sizes used in the article.
Line 34 The author should define what he means by HTA and ‘value assessment’ plus the role of the NPC (there is potentially an international audience). The author used the term HTA but I wonder if he really means economic evaluations and, more specifically, cost utility analyses. HTA encompasses aspects well beyond a cost-utility analysis. The WHO also supports HTA (not just ISPOR and ICER): https://www.who.int/medical_devices/assessment/en/
Response: good point I have clarified with insert. Lines 23-25.
Also: check the title it refers to the construction of imaginary worlds in health technology assessment
In recent months, many governments have relied on modelling to make decisions about how to manage the COVID-19 pandemic. Would the author argue that such ‘imaginary world’ modelling is also ‘pseudoscience’?
Response: no because in pandemic modelling we are dealing with short term and modifiable forecasts. In the application of cost-per-QALY these models are typically for the lifetime of a target patient group. In the covid case these are forecasts that can be empirically assessed; in the lifetime model there is no hope of evaluation of claims. Hence the term used
Line 42 I am not sure of their reference to the ‘Dover courtroom’. Needs a citation.
Response: citation included
Line 90. Presumably ICER here means the organisation and not the metric.
Response: see line 31 where ICER is spelled out
Line 124 define OED
Response : Have changed to Oxford English Dictionary
Line 27. The author proposes a ‘way out’ but the actual implementation of the proposal is not clear. How would the RMT be manifested in practice?
Response: I have added A number of these have been developed following RMT standards. One example is the Crohn’s Life Impact Questionnaire (CLIQ). Lines 255-6 with citations
Reviewer 2 Report
The contributions is made in a form of a commentary, were the author provides his opinion on the Approach of the National Pharmaceutical Council to Value Assessment.
The comment is well structured and well developed.
Although several argumentations are provided, I miss the identification of some sources to backup claims and make the contribution stronger. In other to better guide the reader and to make the argumentation more transparent, some claims should be supported.
I have included some minor comments in the text.
Some sentences are difficult to read due to the lack of punctuation. A minor revision is advice.
Author Response
REVIEWER 2
The contributions is made in a form of a commentary, were the author provides his opinion on the Approach of the National Pharmaceutical Council to Value Assessment.
The comment is well structured and well developed.
Although several arguments are provided, I miss the identification of some sources to backup claims and make the contribution stronger. In other to better guide the reader and to make the argumentation more transparent, some claims should be supported.
I have included some minor comments in the text.
Some sentences are difficult to read due to the lack of punctuation. A minor revision is advice.
Response
I have added additional references ,and read through the text to check out punctuation etc.
Reviewer 3 Report
This commentary is about the Author's theoretical criticism on the recently recommended Health Technology Assessment standards of the National Pharmaceutical Council in the US. Specifically, the Author re-expresses his misesteem against health economic modelling ("fabricating imaginary world", "pseudoscience", "approximate (dis)information", "unevaluable results", "science fiction literature", to mention a few roundabout phrases, as opposed with "normal science" and "discovery of new facts". Many citations in the paper go back to decades or even centuries when arguing on research theory, which is intellectually appealing. On the other hand, it is disappointing that the commentary does not intend to give a balanced assessment of the new HTA standards, and almost completely fail to explain the advantages of the proposed NPC HTA methods. Nevertheless, it is briefly mentioned in the paper that health economic modelling (i.e. "constructing imaginary evidence") is "a lot easier and more cost-effective" than conducting appropriately designed and conducted clinical studies with 10-20-30 years of patient follow-up, to answer the same questions with direct observable evidence. Note that even if such studies would be sponsored and initiated, their results will be available only after decades, therefore, clinical data collection is not a real alternative to health economic modelling in several, more urgent decision contexts. The Author also argues against generic health utility measurements (on the grounds that the lowest possible utility value in EQ-5D-3L and EQ-5D-5L is negative, and these scales integrate multiple symptom domains into a single QALY score).
To put this paper into broader context, first it is important to understand that Health Technology Assessment is not a sterile science, but is a pragmatic approach to ensure better allocative efficiency, transparency, and reproducibility in health policy decisions. Transparency and reproducibility are ensured by the reported details of the model assumptions and structures. To compare the allocative efficiency across different medical fields, the net health benefits must be translated into a common metric - which is typically QALY in health technology assessment (and DALY in food safety benefit-risk assessment). Decisions made without health economic modelling will not be better in any way. The Author fails to explain how the justifiable price of new medical technologies could be identified via "normal science" - which was neither the case before the adoption of the NPC HTA standards. Instead, pricing negotiations of health technologies in the US were mostly subject to market mechanisms, with probably higher risk of insufficient efficiency from the public perspective, lower transparency, and questionable reproducibility. Health economic models make the decision process more explicit, reproducible, and accountable. The recently introduced NPC HTA standards in the US are very similar to HTA standards and practices well established in several EU countries. Notably, health systems in the latter countries operate more efficiently, yielding similar or even higher health benefits to the society at lower costs as compared with the US. Adopting the new NPC HTA standards in the US can be seen as a promising research experiment, and it will hopefully generate directly observable evidence on improved health system sustainability and performance. On the other hand, the new standards may imply lower justifiable prices for new medical technologies in the largest pharmaceutical market of the world. Therefore, to confirm that the expressed opinions are authentic, the Author is invited to declare any competing interests.
I suggest reformulating the below sentence (Lines 128-130): "Does imaginary evidence (or claims) constructed from lifetime models trump (apologies!) evidence based medicine...". Making word joke / pun on the family name of the US President in a scientific manuscript does not add to the arguments and is not to be endorsed, as far as I can see it from the EU.
As a major revision, I suggest providing a more balanced evaluation of the NPC HTA standards, and explaining the proposed alternative: how the justifiable price level of new health technologies could be identified at the time of market launch via "normal science".
Author Response
REVIEWER 3
A major rewrite is not necessary as the reviewer missed the point that the commentary was directed to the NPC’s position on the construction of lifetime imaginary worlds. It is for this reason that the standards of normal science and the impossible nature of QALYs is central to the argument. The review also seems unaware of the constraints imposed by the axioms of fundamental measurement on imaginary QALY constructs.
I have made two changes: lines 128-30 have been redrafted to remove the pun; and I have added lines 135-37 to make quite clear that the answer is not to wait decades for suitable evidence. I have added a reference to a paper I wrote in which gives this advice and to a more recent paper on guidelines I developed for the University of Minnesota..
Round 2
Reviewer 3 Report
I disagree with the Author in many points, as explained in my previous review comments. The provided brief response does not elucidate how the Author envisages the decision making process: how the justifiable price level of new health technologies could be identified and agreed at the time of market launch via "normal science". Now the Author proposes judging the fair price based on a study protocol - a protocol which does not include directly observable evidence but only research plans for the "imaginary" future. I cannot see in which instance could a study protocol allow a better estimate of justified value for a new medical technology than a health economic model. Or maybe the Author is restricting the value proposition to those outcomes that can be measured in very short-term studies, to avoid any significant delay of the price negotiations and launch? These questions remain unanswered. Formulary committees without the philosopher's stone will not be in a better position if they have to rely on study protocols. In the revised version, additional reference is made to previous papers of the Author, suggesting in the response document that those papers explained better the proposed approach. It would be nice if the relevant recommendations would be briefly summarized also in this paper. Decreasing the redundancy in the criticism would allow some space for this purpose, if necessary.
Moreover, the "mathematically impossible" QALYs are easy to calculate without violating any rule of mathematics. Note that QALY values are never divided by each other, but the QALY difference is always considered, therefore the starting point of the scale (whether starting at zero or at a negative value) has no practical relevance. And health economics is a practice-oriented, pragmatic research field. When calculating the incremental cost-effectiveness ratio, the difference in costs is divided by the difference in QALYs, comparing the new technology to the most relevant alternative comparator (e.g. current standard of care).
The revised manuscript still does not include the conflicting interests declaration of the Author. I see no reason for this weakness and insist to add it to the paper. Given the numerous papers of the Author criticizing the health economic modelling and health benefit quantification approaches, this nuance is clearly relevant, and the lack of the declaration would raise concerns regarding the motives of the presented opinions.
Based on our previous correspondence, I do not think that our viewpoints will converge. As minor revision, I recommend supplementing the paper with the conflicting interests declaration of the Author.
Author Response
I disagree with the Author in many points, as explained in my previous review comments. The provided brief response does not elucidate how the Author envisages the decision making process: how the justifiable price level of new health technologies could be identified and agreed at the time of market launch via "normal science". Now the Author proposes judging the fair price based on a study protocol - a protocol which does not include directly observable evidence but only research plans for the "imaginary" future. I cannot see in which instance could a study protocol allow a better estimate of justified value for a new medical technology than a health economic model. Or maybe the Author is restricting the value proposition to those outcomes that can be measured in very short-term studies, to avoid any significant delay of the price negotiations and launch? These questions remain unanswered. Formulary committees without the philosopher's stone will not be in a better position if they have to rely on study protocols. In the revised version, additional reference is made to previous papers of the Author, suggesting in the response document that those papers explained better the proposed approach. It would be nice if the relevant recommendations would be briefly summarized also in this paper. Decreasing the redundancy in the criticism would allow some space for this purpose, if necessary.
REPLY
The reviewer obviously does not understand the standards for normal science: claims must be credible, evaluable and replicable. Building an imaginary framework that goes 30 years into the future to support value assessment is pseudoscience; bunk. There is no such thing as justified value. Formulary committees look to the available evidence for provisional acceptance and to support pricing – it is their decision. If this is the philosopher’s stone (good phrase; I shall have to use it in future critiques). Let’s stay with real world not imaginary evidence.
Moreover, the "mathematically impossible" QALYs are easy to calculate without violating any rule of mathematics. Note that QALY values are never divided by each other, but the QALY difference is always considered, therefore the starting point of the scale (whether starting at zero or at a negative value) has no practical relevance. And health economics is a practice-oriented, pragmatic research field. When calculating the incremental cost-effectiveness ratio, the differencein QALYs is divided by the difference in costs, comparing the new technology to the most relevant alternative comparator (e.g. current standard of care).
REPLY
This is nonsense. The utility score is an ordinal measure (perhaps he/she might check the internet – try “Measurement Scales’). Why: (i) it is based in case of EQ-5D-3L and other multiattribute scales on ordinal response4s and (ii) there was no attempt to build a scale that had ratio properties. The utilities fail to have a true zero. You can’t talk about differences in utilities (or imaginary QALYs) because the utility is on an ordinal scale – ranking but no differences. The QALY IS AN IMPOSSIBLE CONSTRUCT BECAUSE THE UTILITY IS ORDINAL AND WILL NOT SUPPORT CREATING A QALY AS YOU CANNOT MULTIPLY TIME SPENT BY AN ORDINAL SCORE. Obviously, if after 30 years of promoting imaginary and mathematically impossible constructs you feel a need to protect this type of technology assessment, then this would be a standard yet false response. LET ME REPEAT: THE UTILITY SCALE IS ORDINAL. IT WILL NOT SUPPORT ANY MATEMATICAL OPERATION: ADDITION, SUBTRACTION, MULTIPLICATION OR DIVISION. TALKING ABOUT QALY DIFFERENCES WHEN THE QALY IS A MATHEMATICALLY IMPOSSUIBLE CONSTRUCT IS JUST NONSENSE.
The revised manuscript still does not include the conflicting interests declaration of the Author. I see no reason for this weakness and insist to add it to the paper. Given the numerous papers of the Author criticizing the health economic modelling and health benefit quantification approaches, this nuance is clearly relevant, and the lack of the declaration would raise concerns regarding the motives of the presented opinions.
REPLY
There is no conflict of interest. As to previous work (some 30 plus peer reviewed papers/commentaries, this has made quite clear my position.
Based on our previous correspondence, I do not think that our viewpoints will converge. I agree with publication as long as the reviewer comments are also published together with the paper, to allow a balanced view for the readers. As minor revision, I recommend supplementing the paper with the conflicting interests declaration of the Author.
REPLY
Please: I would really appreciate these comments to be printed. This message must reach as wide an audience as possible. It might encourage the audience to understand the axioms of fundamental measurement. You might want to attach a list of all my commentaries/papers on the failure of reference case models that have been published over the past 5 years..